# Taking Apart Autoencoders: How do They Encode Geometric Shapes ?

## Abstract

We study the precise mechanisms which allow autoencoders to encode and decode a simple geometric shape, the disk. In this carefully controlled setting, we are able to describe the specific form of the optimal solution to the minimisation problem of the training step. We show that the autoencoder indeed approximates this solution during training. Secondly, we identify a clear failure in the generalisation capacity of the autoencoder, namely its inability to interpolate data. Finally, we explore several regularisation schemes to resolve the generalisation problem. Given the great attention that has been recently given to the generative capacity of neural networks, we believe that studying in depth simple geometric cases sheds some light on the generation process and can provide a minimal requirement experimental setup for more complex architectures.

## 1 Introduction

Autoencoders are neural networks, often convolutional neural networks, whose purpose is twofold. Firstly, to compress some input data by transforming it from the input domain to another space, known as the latent, or code, space. The second goal of the autoencoder is to take this latent representation and transform it back to the original space, such that the output is similar, with respect to some criterion, to the input. One of the main objectives of this learning process being to reveal important structure in the data via the latent space, and therefore to represent this data in a more meaningful fashion or one that is easier to model. Autoencoders have been proven to be extremely useful in many tasks ranging from image compression to synthesis. Many variants on the basic idea of autoencoders have been proposed, the common theme being how to impose useful properties on the learned latent space. However, very little is known about the actual inner workings and mechanisms of the autoencoder.

The goal of this work is to investigate these mechanisms and describe how the autoencoder functions. Many applications of autoencoders or similar networks consider relatively high-level input objects, ranging from the MNIST handwritten digits to abstract sketches of conceptual objects (Zhu et al. (2016); Ha & Eck (2017)). Here, we take a radically different approach. We consider, in depth, the encoding/decoding processes of a simple geometric shape, the disk, and investigate how the autoencoder functions in this case. There are several important advantages to such an approach. Firstly, since the class of objects we consider has an explicit parametrisation, it is possible to describe the "optimal" performance of the autoencoder, ie. can it compress and uncompress a disk to and from a code space of dimensionality 1 ? Secondly, the setting of this study fixes certain architecture characteristics of the network, such as the number of layers, leaving fewer free parameters to tune. This means that the conclusions which we obtain are more likely to be robust than in the case of more high-level applications. Finally, it is easier to identify the roles of different components of the network, which enables us to carry out an instructive ablation study.

Using this approach, we show that the autoencoder approximates the theoretical solution of the training problem when no biases are involved in the network. Secondly, we identify certain limitations in the generalisation capacity of autoencoders when the training database is incomplete with respect to the underlying manifold. We observe the same limitation using the architecture of Zhu et al. (2016), which is considerably more complex and is proposed to encode natural images. Finally, we analyse several regularisation schemes and identify one in particular which greatly aids in overcoming this generalisation problem.

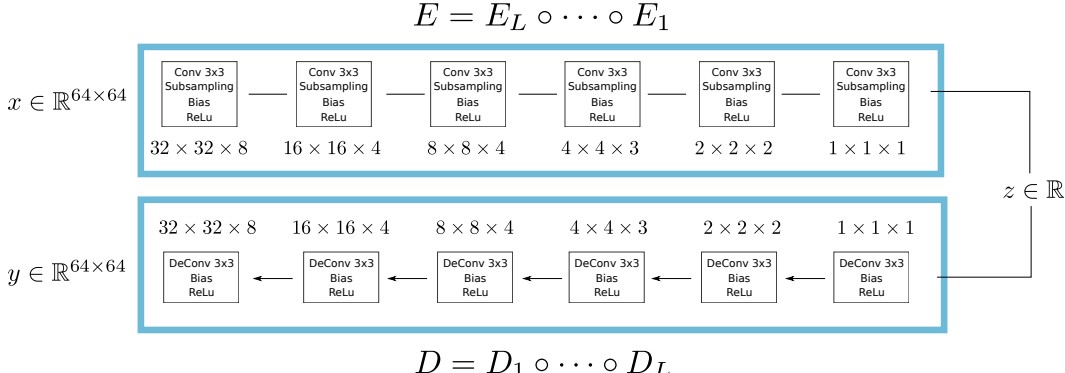

Figure 1: Generic autoencoder architecture used in the geometric experiments.

## 2    PRIOR WORK

The concept of autoencoders has been present for some time in the learning community (LeCun (1987); Bourlard & Kamp (1988)). The objective is to train two networks, an "encoder" and a "decoder", which transform the input data to and from a code, or latent, space which is learned by the algorithm. In many applications, the dimensionality $d$ of the latent space is smaller than that of the original data, so that the autoencoder is encouraged to discover useful features of the data. In practice, we obviously do not know the exact value of $d$, but we would still like to impose as much structure in the latent space as possible. This idea lead to the regularisation in the latent space of autoencoders, which comes in several flavours. The first is the sparse autoencoder (Ranzato et al. (2007)), which attempts to have as few active (non-zero) neurons as possible in the network. This can be done either by modifying the loss function to include sparsity-inducing penalisations, or by acting directly on the values of the code $z$. In the latter option, one can use rectified linear units (ReLUs) to encourage zeros in the code (Glorot et al. (2011)) or simply specifying a maximum number of non-zero values as in the "k-sparse" autoencoder (Makhzani & Frey (2013)). Another approach, taken by the variational autoencoder, is to specifying the a priori distribution of the code $z$. Kingma & Welling (2014) use the Kullback-Leibler divergence to achieve this goal, and the authors suppose a Gaussian distribution of $z$. The "contractive" autoencoder (Rifai et al. (2011)) encourages the derivatives of the code with respect to the input image to be small, meaning that the representation of the image should be robust to small changes in the input.

Autoencoders can be applied to a variety of problems, such as denoising ("denoising autoencoder") or image compression (Ballé et al. (2016)). For a good overview of autoencoders, see the book of Goodfellow et al. (Goodfellow et al. (2016)). Recently, a great deal of attention has been given to the capacity of CNNs, and in particular generative adversarial networks (GANs) (Radford et al. (2015)) or autoencoders, to generate new images. It is well-known that these networks have important limitations, such as the tendency to produce low quality images or to reproduce images from the training set because of mode collapse. But despite these limitations, many works have investigated the generative capacity of such networks, see for instance Dosovitskiy & Brox (2016); Salimans et al. (2016); Reed et al. (2016); Zhu et al. (2016) and often demonstrated intriguing visual results. In this context, a natural question is : how efficient are such networks at inventing realistic new images ? How well do they generalize visual content ?

## 3    HOW DO AUTOENCODERS PROCESS VISUAL IMAGES ?

Although autoencoders have been extensively studied, very little is known concerning the actual inner mechanics of these networks, in other words quite simply, how they work. This is obviously much too vast a question in the general case, however very often deep learning is applied to the specific case of *images*. In this work, we aim to discover how, with a cascade of simple operations common in deep networks, an autoencoder can encode and decode very simple images. In view of this goal, we propose to study in depth the case of *disks* of variable radii. This controlled setting and

| Layer | Input | Hidden layers | | | | Code ($z$) |
|-------|-------|-------|-------|-------|-------|-------|
| Depths | 1 | 8 | 4 | 4 | 3 | 2 | 1 |

| Parameter | Spatial filter size | Non-linearity | Learning rate | Learning algorithm | Batch size |
|-----------|---------------------|---------------|---------------|--------------------|------------|
| Value | $3 \times 3$ | Leaky ReLu ($\alpha = 0.2$, see Eq. (2)) | 0.001 | Adam | 300 |

Table 1: Parameters of autoencoder designed for processing centred disks of random radii.

careful study of the autoencoder are the main goals of the paper, and structure our work throughout. Before continuing, we describe our autoencoder in a more formal fashion.

## 3.1 NOTATION AND AUTOENCODER ARCHITECTURE

We denote input images with $x \in \mathbb{R}^{m \times n}$ and $z \in \mathbb{R}^d$, where $m$ and $n$ are the height and the width of the image, respectively, and $d$ is the dimension of $z$. The autoencoder consists of the couple $(E, D)$, the encoder and decoder which transform to and from the "code" space, with $E : \mathbb{R}^{m \times n} \to \mathbb{R}^d$ and $D : \mathbb{R}^d \to \mathbb{R}^{m \times n}$. As mentioned, the goal of the auto-encoder is to compress and uncompress a signal into a representation with a smaller dimensionality, while losing as little information as possible. Thus, we search for the parameters of the encoder and the decoder, which we denote with $\Theta_E$ and $\Theta_D$ respectively, by minimising

$$(\Theta_E, \Theta_D) = \operatorname*{argmin}_{\Theta_E, \Theta_D} \sum_x ||x - D(E(x))||_2^2 \tag{1}$$

The autoencoder consists of a series of convolutions with filters of small compact support, sub-sampling/up-sampling, biases and non-linearities. The values of the filters are termed the weights of the network, and we denote the encoding filters with $w_{\ell,i}$, where $\ell$ is the layer number and $i$ the number of the filter. Similarly, we denote the decoding filters $w'_{\ell,i}$, the encoding and decoding biases $b_{\ell,i}$ and $b'_{\ell,i}$. We choose leaky ReLUs for the non-linearities :

$$\phi_\alpha(x) = \begin{cases} x, & \text{for } x \geq 0 \\ \alpha x, & \text{for } x < 0 \end{cases}, \tag{2}$$

with parameter $\alpha = 0.2$. Thus, the output of a given encoding layer is given by

$$E_i^{l+1} = \phi_\alpha(E^l * w_{\ell,i} + b_{\ell,i}), \tag{3}$$

and similarly for the decoding layers (except for an zero-padding upsampling prior to the convolution) , with weights and biases $w'$ and $b'$, respectively.

We consider images of a fixed (square) spatial support $\Omega = [0, m-1] \times [0, m-1]$ and also that the subsampling rate $s$ is fixed. In the encoder, subsampling is carried out until and $z$ is a single scalar. Thus, the number of layers in our encoder and decoder is not an independent parameter. We set the support of all the convolutional filters in our network to $3 \times 3$. The architecture of our autoencoder remains the same throughout the paper, and is shown in Figure 1. We summarise our parameters in Table 1. We now investigate the inner mechanics of autoencoders in the case of a simple geometric shape: the disk.

## 3.2 AUTOENCODING DISKS

Our training set consists of binary images of centred disks of random radii, with one disk per image in the test database. Each disk image is determined by the indicator function of a disk of radius $r$, and is therefore binary. Theoretically, an optimal encoder would only need one scalar to represent the image. Therefore the architecture in Figure 1 is set up to ensure a code size $d = 1$. Our first important observation (see Figure 2) is that not only can the network learn to encode/decode

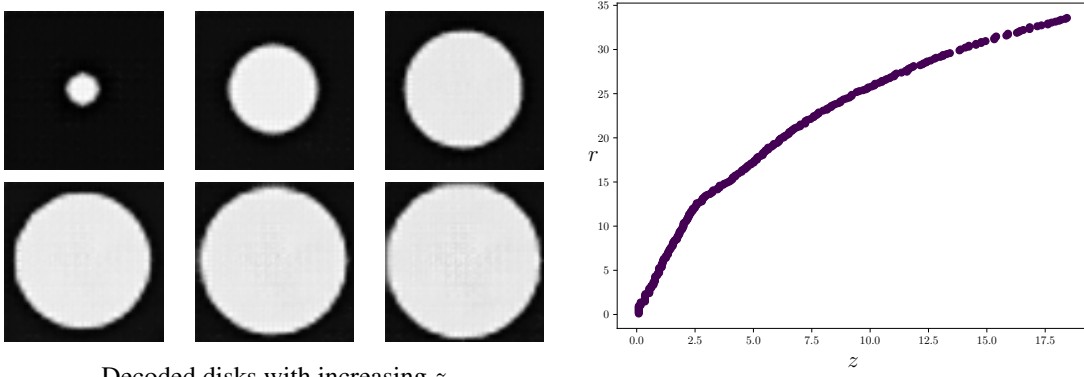

Decoded disks with increasing $z$

Figure 2: **Investigating the latent space in the case of disks**. On the left side, we have interpolated $z$ in the latent space between two encoded input disks (one small and one large), and show the decoded, output image. It can be seen that the training works well, with the resulting code space being meaningful. On the right, we plot the radii of the input disks against their codes $z \in \mathbb{R}$. The autoencoder appears to represent the disks with their area.

disks, but that the code $z$ which is learned can be interpolated and the corresponding decoding is meaningful. Thus, in this case, the autoencoder is able to encode/decode the data in an optimal fashion. We now proceed to see how the autoencoder actually works on a detailed level, starting with the encoding step.

### 3.2.1   ENCODING A DISK

Encoding a centred disk of a certain radius to a scalar $z$ can be done in several ways, the most intuitive being integrating over the *area* of the disk (encoding a scalar proportionate to its area) or integrating over the *perimeter* of the disk (encoding a scalar proportionate to its radius). The empirical evidence given by our experiments points towards the first option, since $z$ seems to represent the area and not the radius of the input disks (see Figure 2). If this is the case, the integration operation can be done by means of a simple cascade of linear filters. As such, we should be able to encode the disks with a network containing only convolutions and sub-sampling, and no having non-linearities. We have verified experimentally this with such an encoder.

### 3.2.2   DECODING A DISK

A more difficult question is how does the autoencoder convert a scalar, $z$, to an output disk of a certain size (the decoding process). One approach to understanding the inner workings of autoencoders, and indeed any neural network, is to remove certain elements of the network and to see how it responds, otherwise known as an *ablation* study. We found that removing the *biases* of the autoencoder leads to very interesting observations. While, as we have shown, the encoder is perfectly able to function without these biases, this is not the case for the decoder. Figure 3 shows the results of this ablation. The decoder learns to spread the energy of $z$ in the output according to a certain function $g$. Thus, the goal of the biases is to shift the intermediary (hidden layer) images such that a cut-off can be carried out to create a satisfactory decoding. We have investigated the behaviour of the decoder without biases in detail. In particular, we will derive an explicit form for the energy minimized by the network, for which a closed form solution can be found (see Appendix A), but more importantly for which we will show experimentally that the network finds the right solution. We first make a general observation about this configuration (without biases).

**Proposition 1.** *[Positive Multiplicative Action of the Decoder Without Bias]*

*Consider a decoder, without biases $D(z) = D^L \circ \cdots \circ D^1(z)$, with $D^{\ell+1} = \phi_\alpha \left( U(D^\ell) * w'_{\ell_i} \right)$, where $U$ stands for upsampling with zero-padding. In this case, the decoder acts multiplicatively on $z$, meaning that*

$$\forall z, \; \forall \lambda \in \mathbb{R}^+, \; D(\lambda z) = \lambda D(z).$$

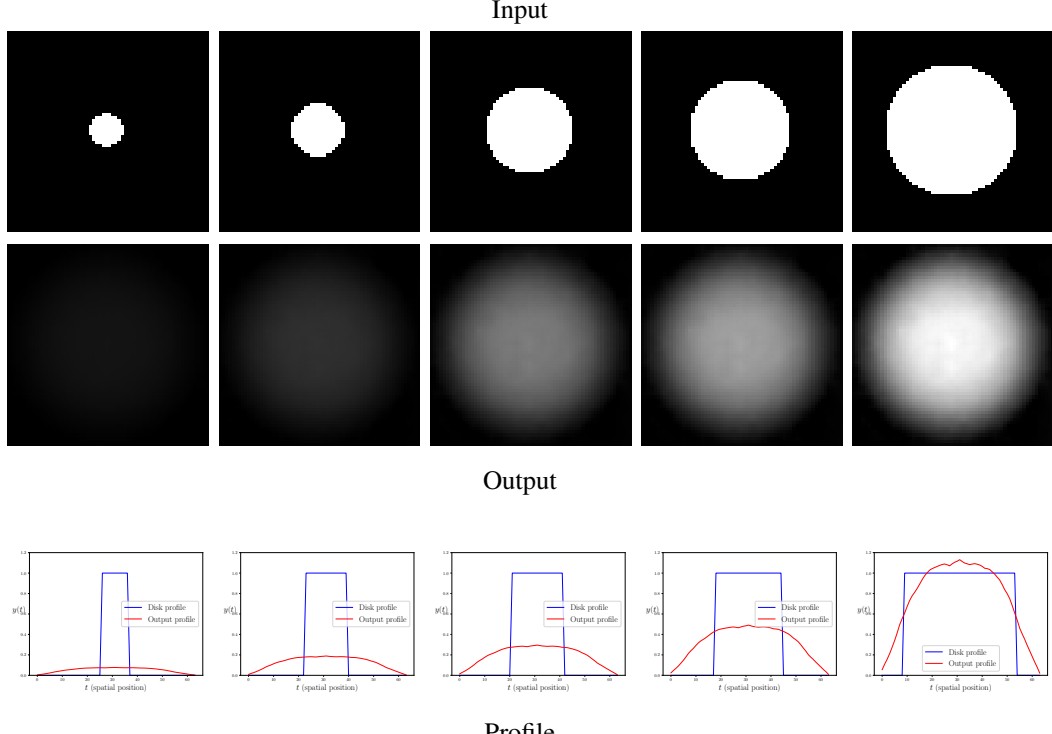

Figure 3: **Autoencoding of disks when the autoencoder is trained with no bias.** The autoencoder learns a function $f$ which is multiplied by a constant scalar, $h(r)$, for each radius. This behaviour is formalised in Equation (5).

*Proof.* For a fixed $z$ and for any $\lambda > 0$. We have

$$
\begin{aligned}
D^1(\lambda z) &= \phi_\alpha \left( U(\lambda z) * w'_\ell \right) \\
&= \max \left( \lambda (U(z) * w'_\ell), 0 \right) + \alpha \min \left( \lambda (U(z) * w'_\ell), 0 \right) \\
&= \lambda \max \left( U(z) * w'_\ell, 0 \right) + \lambda \alpha \min \left( U(z) * w'_\ell, 0 \right) = \lambda \phi_\alpha \left( U(z) * w'_\ell \right) = \lambda D^1(z). \quad (4)
\end{aligned}
$$

This reasoning can be applied successively to each layer up to the output $y$. When the code $z$ is one dimensional, the decoder can be summarized as two linear functions, one for positive codes and a second one for the negative codes. However, in all our experiments, the autoencoder without bias has chosen to use only one possible sign for the code, resulting in a linear decoder. $\square$

Furthermore, the profiles in Figure 3 suggest that a single function is learned, and that this function is multiplied by a factor which is constant for each radius. In light of Proposition 1, this means that the decoder has chosen a fixed sign for the code and that the decoder is linear. This can be expressed as

$$
y(t, r) = h(r) f(t), \quad (5)
$$

where $t$ is a spatial variable and $r \in (0, \frac{m}{2}]$ is the radius of the disk. This is checked experimentally in Figure 7 in Appendix A. In this case, we can write the optimisation problem of the decoder as

$$
\hat{f}, \hat{h} = \operatorname*{argmin}_{f, h} \int_0^R \int_\Omega \left( h(r) f(t) - \mathbb{1}_{B_r}(t) \right)^2 dt \, dr, \quad (6)
$$

where $R$ is the maximum radius observed in the training set, $\Omega = [0, m - 1] \times [0, m - 1]$ is the image domain, and $B_r$ is the disk of radius $r$. Note that we have expressed the minimisation problem for continuous functions $f$. This is not strictly the case, especially for images of small disk radii, however for our purposes the approximation is good. In this case, we have the following proposition.

**Proposition 2** (Decoding Energy for an autoencoder without Biases). *The decoding training problem of the autoencoder without biases has an optimal solution $\hat{f}$ that is radially symmetric and*

Input

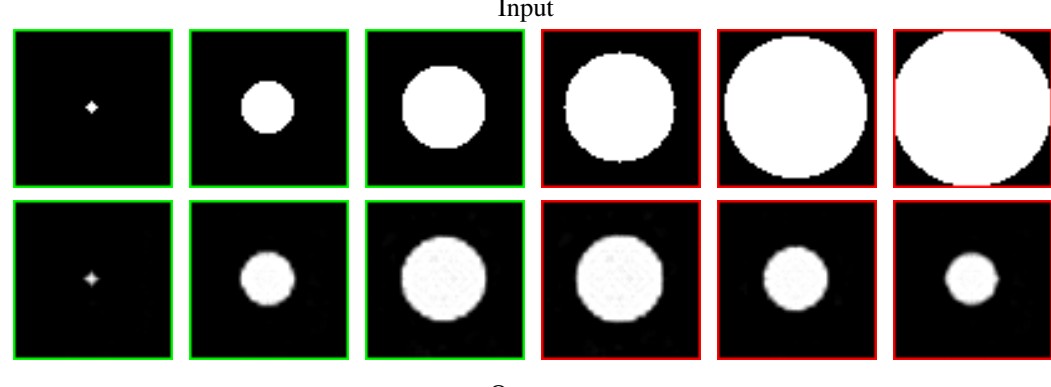

Output

Figure 4: **Autoencoding of disks with a database with limited radii.** The autoencoder is not able to extrapolate further than the largest observed radius. The images with a green border represent disks whose radii have been observed during training, while those in red have not been observed.

*maximises the following energy:*

$$\int_0^R \left( \int_0^r f(\rho) \mathbb{1}_{[0,r]}(\rho) \, \rho \, d\rho \right)^2 \, dr =: E(f), \tag{7}$$

*under the (arbitrary) normalization* $\|f\|_2^2 = 1$.

*Proof.* When $f$ is fixed, the optimal $h$ for Equation (6) is given by

$$\hat{h}(r) = \frac{\langle f, \mathbb{1}_{B_r} \rangle}{\|f\|_2^2}, \tag{8}$$

where $\langle f, \mathbb{1}_{B_r} \rangle = \int_\Omega f(t) \mathbb{1}_{B_r}(t) \, dt$. After replacing this in Equation (6), we find that

$$\hat{f} = \underset{f}{\operatorname{argmin}} \int_0^R -\frac{\langle f, \mathbb{1}_{B_r} \rangle^2}{\|f\|^2} dr = \underset{f}{\operatorname{argmin}} \int_0^R -\langle f, \mathbb{1}_{B_r} \rangle_2^2 \, dr, \tag{9}$$

where we have chosen the arbitrary normalisation $\|f\|_2^2 = 1$. The form of the last equation shows that the optimal solution is obviously radially symmetric[1]. Therefore, after a change of variables, the energy maximised by the decoder can be written as

$$\int_0^R \left( \int_0^r f(\rho) \mathbb{1}_{[0,r]}(\rho) \, \rho \, d\rho \right)^2 \, dr =: E(f), \tag{10}$$

such that $\|f\|_2^2 = 1$. □

In Appendix A, we compare the numerical solution of this problem with the actual profile learned by the network, yielding a very close match. This result is very interesting, since it shows that the training process has achieved the optimal solution, in spite of the fact that the loss is non convex.

### 3.2.3 GENERALISATION AND REGULARISATION

As we have recalled in Section 2, many works have recently investigated the generative capacity of autoencoders or GANs. Nevertheless, it is not clear that these architectures truly invent or generalize some visual content. A simpler question is : to what extent is the network able to generalise a simple geometric notion ? In this section, we address this issue in our restricted but interpretable case.

---

[1]If not, then consider its mean on every circle, which decreases the $L^2$ norm of $f$ while maintaining the scalar product with any disk. We then can increase back the energy by deviding by this smaller $L^2$ norm according to $\|f\|_2 = 1$.

Input

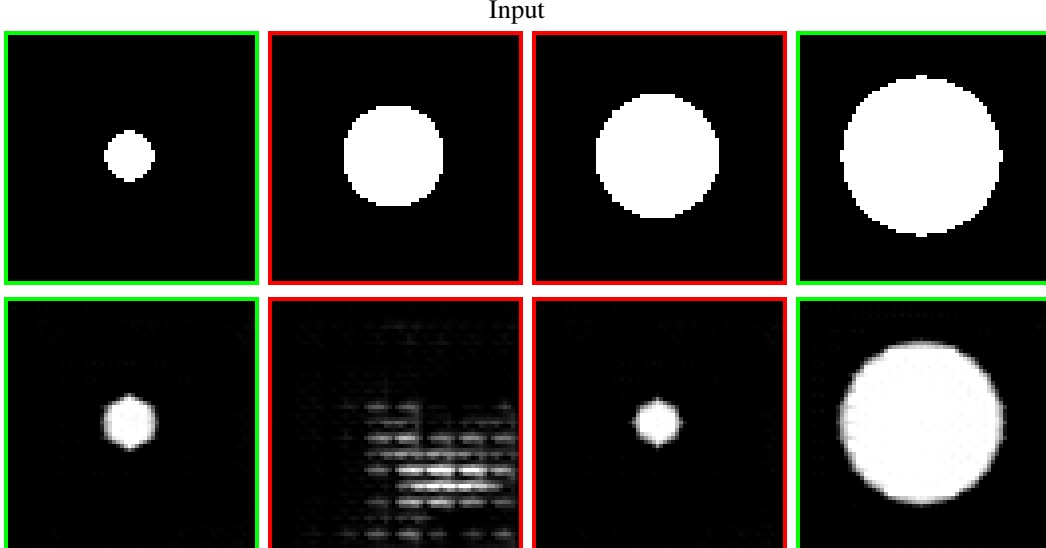

Figure 5: **Input and output of our network when autoencoding examples of disks when the database contains a "hole".** Disks of radii between 11 and 18 pixels (out of 32) were not observed in the database. In green, the disks whose radii have been observed in the database, in red those which have not.

For this, we study the behaviour of our autoencoder when examples are removed from the training dataset. In Figure 4, we show the autoencoder result when the disks with radii above a certain threshold $R$ are removed. The radii of the left three images (with a green border) are present in the training database, whereas the radii of the right three (red border) have not been observed. It is clear that the network lacks the capacity to extrapolate further than this radius. Indeed, the autoencoder seems to project these disks onto smaller, observed, disks, rather than learning the abstraction of a disk.

Again by removing the biases from the network, we may explain why the autoencoder fails to extrapolate when a maximum radius $R$ is imposed. In Appendix B, we show experimental evidence that in this situation, the autoencoder learns a function $f$ whose support is restricted by the value of $R$, leading to the autoencoder's failure. However, a fair criticism of the previous experiment is simply that the network (and deep learning in general) is not designed to work on data which lie outside of the domain observed in the training data set. Nevertheless, it is reasonable to expect the network to be robust to such "holes" *inside* the domain. Therefore, we have also analysed the behaviour of the autoencoder when we removed training datapoints whose disks' radii lie within a certain range, between 11 and 18 pixels (out of a total of 32). We then attempt to reconstruct these points in the test data. Figure 5 shows the results of this experiment. Once again, in the unknown regions the network is unable to recreate the input disks. Goodfellow et al. (2016) (page 521) and Bengio & Monperrus (2005) propose several explanations in the deep learning literature of this phenomenon, such as a high curvature of the underlying data manifold, noisy data or high intrinsic dimensionality of the data. In our setting, *none of these explanations is sufficient*. Thus we conclude that, even in the simple setting of disks, the "classic" autoencoder cannot generalise correctly when a database contains holes.

This behavior is potentially problematic for applications which deal with more complex natural images, lying on a high-dimensional manifold, as these are likely to contain such holes. We have therefore carried out the same experiments using the state-of-the-art "iGAN" approach of Zhu et al. (2016), which is in turn based on the work of Radford et al. (2015), "DCGAN". The visual results of their algorithm are displayed in Appendix C. We trained their network using both a code size of $d = 100$ (as proposed by the authors), and $d = 1$ in order to ensure fair comparisons. Indeed, in our case, not only the dimension of the latent space should be $d = 1$, but also the amount of training data is not enough to work with $d = 100$. Although the $d = 1$ case leads to improved results, in

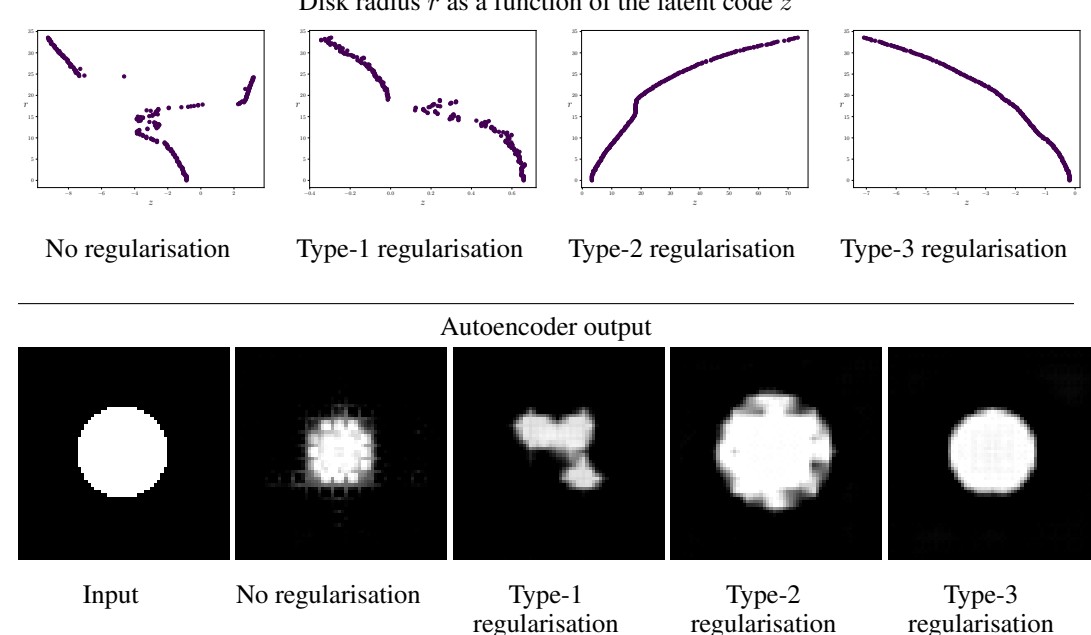

Figure 6: **Result of different types of regularisation on autoencoding in an "unknown region" of the training database.** We have encoded/decoded a disk which was not observed in the training dataset. We show the results of four experiments: no regularisation, $\ell_2$ regularisation in the latent space ("Type 1"), $\ell_2$ weight penalisation of the encoder and decoder ("Type 2") and $\ell_2$ weight penalisation of the encoder only ("Type 3").

both cases the network fails to correctly autoencode the disks belonging to the unobserved region. This shows that the generalisation problem is likely to be ubiquitous, and indeed observed in more sophisticated networks, designed to learn natural images manifolds, even in the simple case of disks. We therefore believe that this issue deserves careful attention. Actually this experiment suggests that the capacity to generate new and simple geometrical shapes could be taken as a minimal requirement for a given architecture.

In order to address the problem, we now investigate several regularisation techniques whose goal is to aid the generalisation capacity of neural networks.

### 3.2.4 REGULARISATION

We would like to impose some structure on the latent space in order to interpolate correctly in the case of missing datapoints. This is often achieved via some sort of regularisation. This regularisation can come in many forms, such as imposing a certain distribution in the latent space, as in variational autoencoders (Kingma & Welling (2014)), or by encouraging $z$ to be sparse, as in sparse auto-encoders (Ranzato et al. (2007); Makhzani & Frey (2013)). In the present case, the former is not particularly useful, since a probabilistic approach will not encourage the latent space to correctly interpolate. The latter regularisation does not apply, since we already have $d = 1$. Another commonly used approach is to impose an $\ell_2$ penalisation of the weights of the filters in the network. The idea behind this bears some similarity to sparse regularisation; we wish for the latent space to be as "simple" as possible, and therefore hope to avoid over-fitting.

We have implemented several regularisation techniques on our network. Firstly, we attempt a simple regularisation of the latent space by requiring a "locality-preservation" property as suggested in Hadsell et al. (2006); Alain & Bengio (2014); Liao et al. (2017), namely that the $\ell_2$ distance between two images $(x, x')$ be maintained in the latent space. This is done by randomly selecting a neighbour of each element in the training batch. Secondly, we regularise the weights of the encoder and/or the

decoder. Thus, our training attempts to minimise the sum of the data term, $\|x - D(E(x))\|_2^2$, and a regularisation term $\lambda\psi(x, \theta)$, which can take one of the following forms:

- Type 1 : $\psi(x, x') = (\|x - x'\|_2^2 - \|E(x) - E(x')\|_2^2)^2$;

- Type 2 : $\psi(\Theta_E, \Theta_D) = \sum_{\ell=1}^{L} \|w_{\cdot,\ell}\|_2^2 + \|w'_{\cdot,\ell}\|_2^2$;

- Type 3 : $\psi(\Theta_E) = \sum_{\ell=1}^{L} \|w_{\cdot,\ell}\|_2^2$;

Figure 6 shows the results of these experiments. First of all, we observe that the type 1 regularisation does not work satisfactorily. One interpretation of this is that the manifold in the training data is "discontinuous", and therefore there are no close neighbours for the disks on the edge of the unobserved region. Therefore, this regularisation is to be avoided in cases where there are significant holes in the sampling of the data manifold. The second type of regularisation, minimising the $\ell_2$ norm of the encoder and decoder weights, produces an interesting effect. Indeed, while the manifold seems reasonable, upon closer inspection, the code $z$ increases in amplitude during the training. Thus, the network cannot converge to a stable solution, which worsens the quality of the results. Finally, we observe that regularising the weights of the encoder works particularly well, and that the resulting manifold is continuous and correctly represents the area of the disks. Consequently, this asymmetrical regularisation approach is to be encouraged in other applications of autoencoders.

At this point, we take the opportunity to note that the clear, marked effects seen with the different regularisation approaches are consistently observed in different training runs. This is due in large part to the controlled, simple setting of autoencoding with disks. Indeed, many other more sophisticated networks, especially GANs, are known to be very difficult to trainSalimans et al. (2016), leading to unstable results or poor reproducibility. We believe that our approach can be of use to more high-level applications, by making it easier to clearly identify which components and regularisations schemes best help in processing complex input data.

### 3.3 CONCLUSION AND FUTURE WORK

We have investigated in detail the specific mechanisms which allow autoencoders to encode image information in an optimal manner in the specific case of disks. We have shown that, in this case, the encoder functions by integrating over disk, and so the code $z$ represents the area of the disk. In the case where the autoencoder is trained with no bias, the decoder learns a single function which is multiplied by scalar depending on the input. We have shown that this function corresponds to the optimal function. The bias is then used to induce a thresholding process applied to ensure the disk is correctly decoded. We have also illustrated certain limitations of the autoencoder with respect to generalisation when datapoints are missing in the training set. This is especially problematic for higher-level applications, whose data have higher intrinsic dimensionality and therefore are more likely to include such "holes". Finally, we identify a regularisation approach which is able to overcome this problem particularly well. This regularisation is asymmetrical as it consists of regularizing the encoder while leaving more freedom to the decoder.

An important future goal is to extend the theoretical analyses obtained to increasingly complex visual objects, in order to understand whether the same mechanisms remain in place. We have experimented with other simple geometric objects such as squares and ellipses, with similar results in an optimal code size. Another question is how the decoder functions with the biases included. This requires a careful study of the different non-linearity activations as the radius increases. Finally, the ultimate goal of these studies is to determine the capacity of autoencoders to encode and generate images representing more complex objects or scenes. As we have seen, the proposed framework can help identifying some limitations of complex networks such as the one from Zhu et al. (2016) and future works should investigate whether this framework can help developing the right regularization scheme or architecture.

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

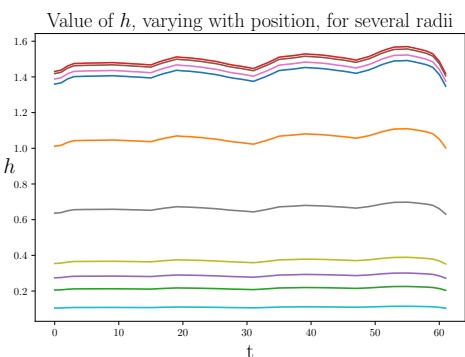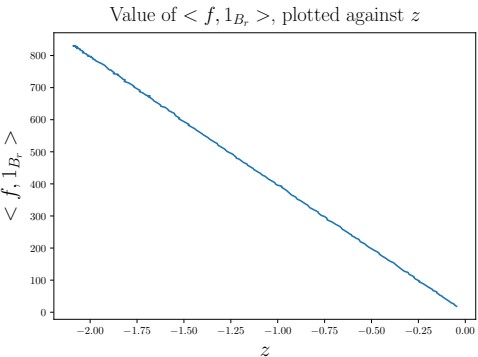

Figure 7: **Verification of the hypothesis that** $y(t, r) = h(r)f(t)$ **for decoding in the case where the autoencoder contains no bias.**. We have determined the average profile of the output of the autoencoder when no biases are involved. On the left, we have divided several random experimental profiles $y$ by the function $h$, and plotted the result, which is close to constant (spatially) for a fixed radius of the input disk. On the right, we plot $z$ against the theoretically optimal value of $h$ ($C \langle f, \mathbb{1}_{B_r} \rangle$, where $C$ is some constant accounting for the arbitrary normalization of $f$). This experimental sanity check confirms our theoretical derivations.

## A  DECODING OF A DISK

During the training of the autoencoder for the case of disks (with no bias in the autoencoder), the objective of the decoder is to convert a scalar into the image of a disk with the $\ell_2$ distance as a metric. Given the profiles of the output of the autoencoder, we have made the hypothesis that the decoder approximates a disk of radius $r$ with a function $y(t; r) = h(r)f(t)$, where $f$ is a continuous function. We show that this is true experimentally in Figure 7 by determining $f$ experimentally by taking the average of all output profiles, and showing the pointwise division of $f$ by randomly selected output profiles. We see that $h$ is approximately constant for varying $t$ and fixed $r$. Please note that we have removed the last spatial coordinate of the profile which suffers from border effects.

We now compare the numerical optimisation of the energy in Equation (7) using a gradient descent approach with the profile obtained by the autoencoder without biases. The resulting comparison can be seen in Figure 8. One can also derive a closed form solution of Equation (7) by means of the Euler-Lagrange equation and see that the optimal $f$ for Equation (7) is the solution of the differential equation $y'' = -kty$ with initial state $(y, y') = (1, 0)$, where $k$ is a free positive constant that accommodates for the position of the first zero of $y$. This gives a closed form of the $f$ in terms of Airy functions.

## B  AUTOENCODING DISKS WITH A DATABASE WITH A LIMITED OBSERVED RADIUS

In Figure 9, we see the grey-levels of the input/output of an autoencoder trained (without biases) on a restricted database, that is to say a database whose disks have a maximum radius $R$ which is smaller than the image width. We have used $R = 18$ for these experiments. We see that the decoder learns a useful function $f$ which only extends to this maximum radius. Beyond this radius, another function is used corresponding to the other sign of codes (see proposition 1) that is not tuned.

## C  AUTOENCODING DISKS WITH THE IGAN ZHU ET AL. (2016)

In Figure 10, we show the autoencoding results of the IGAN network of Zhu et al. We trained their network with a code size of both $z = 100$ and $z = 1$. Although the IGAN works better in the latter case, in both experiments the network fails to correctly autoencode disks in the missing radius region which has not been observed in the training database.

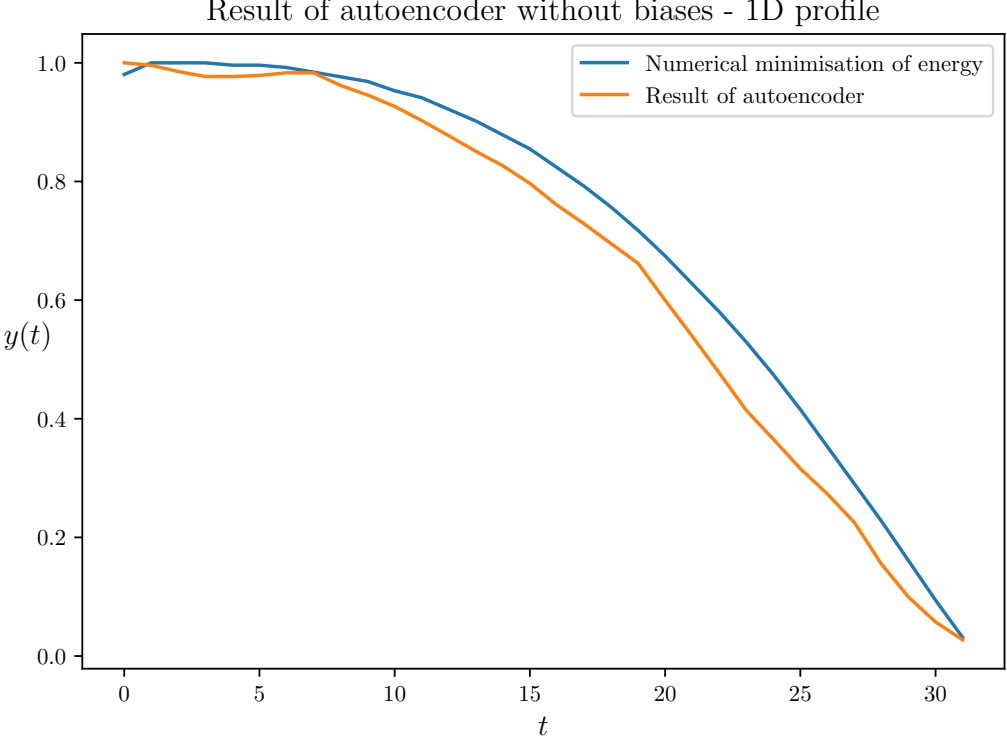

Figure 8: **Comparison of the empirical function $f$ of the autoencoder without biases with the numerical minimisation of Equation** (7)**.** We have determined the empirical function $f$ of the autoencoder and compared it with the minimisation of Equation (7). The resulting profiles are similar, showing that the autoencoder indeed succeeds in minimising this energy.

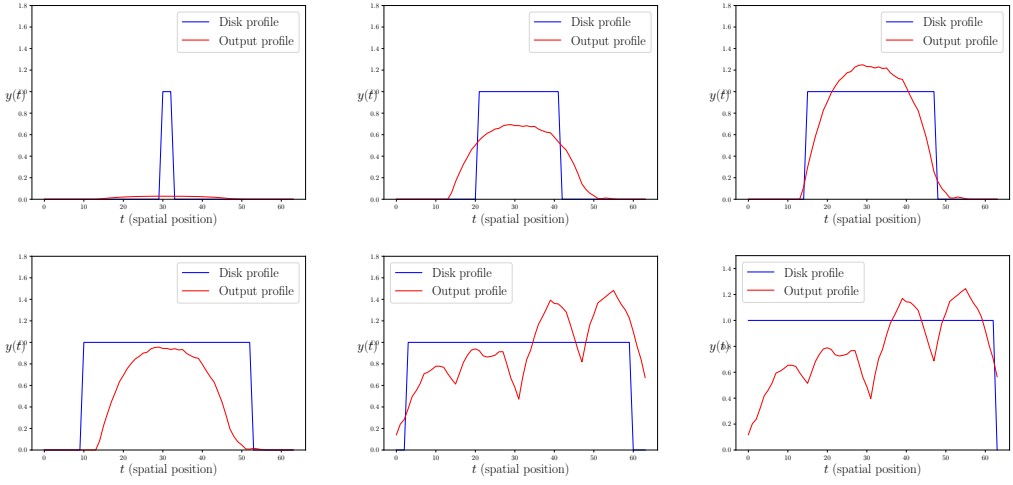

Figure 9: **Profile of the encoding/decoding of centred disks, with a restricted database**. The decoder learns a profile $f$ which only extends to the largest observed radius $R = 18$. Beyond this radius, another profile is learned that has is obviously not tuned to any data.

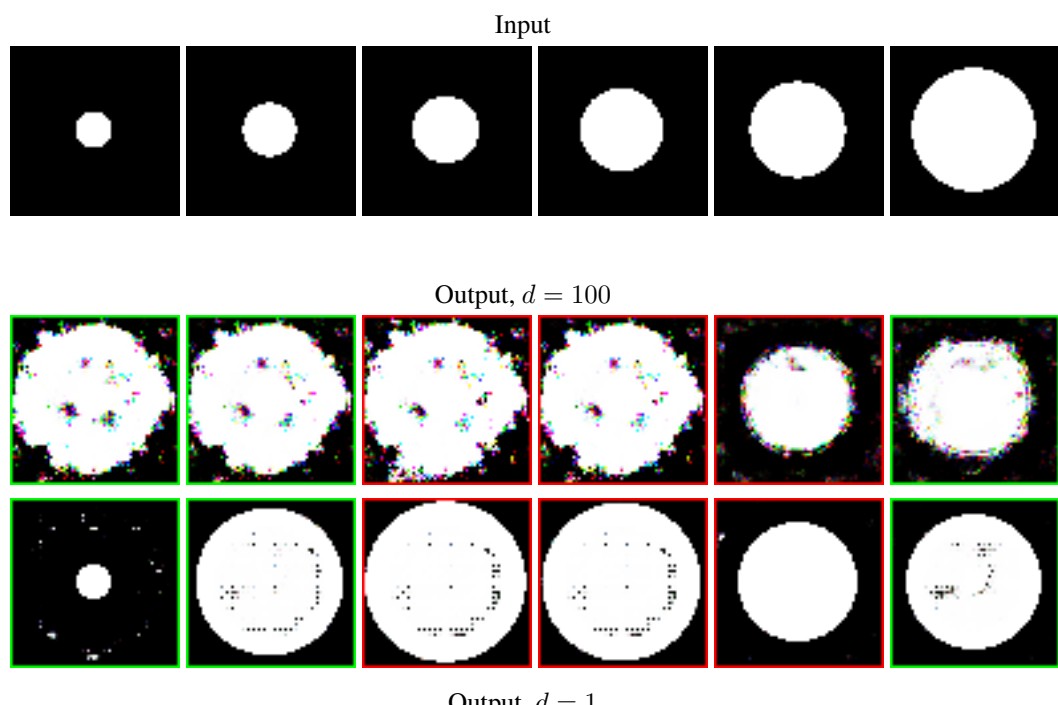

Figure 10: **Input and output of the network of Zhu et al.Zhu et al. (2016) ("IGAN") for disks when the database is missing disks of certain radii.** We have applied the IGAN with a code size of $d = 100$, as in the original paper, and $d = 1$ as in our autoencoder. In both cases the IGAN interpolates incorrectly in the unknown region. Outlined in green are the images with observed radii and in red the unobserved radii.

