# OpenReview forum: "Taking Apart Autoencoders: How do They Encode Geometric Shapes ?"
_ICLR.cc/2018/Conference — Reject_

### Official Review · AnonReviewer3 · 2017-11-26
**This is an interesting study to understand the innerworkings of an autoencoder, however, the study is not quite convncing, yet.**

**Rating:** 4
**Confidence:** 5

**Review:**

1. The idea is interesting, but the study is not comprehensive yet
2. need to visualize the input data space, with the training data, test data, the 'gaps' in training data [see a recent related paper - Stoecklein et al. Deep Learning for Flow Sculpting: Insights into Efficient Learning using Scientific Simulation Data. Scientific Reports 7, Article number: 46368 (2017).].
3. What's the effect of training data size?
4. How do the intermediate feature maps look like?
5. Is there an effect of number of layers? Maybe the network architecture is too deep for the simple data characteristics and size of training set.
6. Other shapes are said to be part of future work, but I am not convinced that serious conclusions can be drawn from this study only?
7. What about the possible effects of Batch normalization and dropout?
8. size of 'd' is critical for autoencoders, only one example in appendix does not do justice, also it seems other color channels show up in the results (fig 10), wasn't it binary input?

---

> ### Author Response · Authors · 2017-12-15
> **Reply to reviewer 3**
>
> 3/ What's the effect of training data size?
>
> In this situation, the training data size is imposed; it is the total possible number of centred disks in a 64 x 64 image.
>
> 4/ How do the intermediate feature maps look like?
>
> In our case, the feature maps resemble disks in the encoding and decoding, which is natural. However we were not able to interpret them in a meaningful way. Thus, we turned to a different approach, the ablation study.
>
> 5/ Is there an effect of number of layers? Maybe the network architecture is too deep for the simple data characteristics and size of training set.
>
> In our case, the number of layers is imposed by the problem and the subsampling coefficient (1/2). We experimented with other, more drastic subsampling, with less success; we chose the minimal architecure which worked correctly.
>
> 6/ Other shapes are said to be part of future work, but I am not convinced that serious conclusions can be drawn from this study only?
>
> We have added some further experiments on ellipses in the supplementary material
> https://www.dropbox.com/s/hn2akqgqh9m0qxg/autoencoders_sup_mat.pdf?dl=0
>
> 7/ What about the possible effects of Batch normalization and dropout?
>
> Batch normalisation is not explored in our case, it is true . In the first part of the paper, we are concerned with an optimal theoretical decoding solution, and we show that the network without biases is able to find this solution, so batch normalisation would not help here. In the second part, we are concerned with improving robustness to missing data. This can only be remedied by regularisation; batch normalisation may speed up convergence, but if it is to an incorrect solution, it is not that useful. Dropout can improve the generalisation capacities of the network, this is true, but only when the network has an excessive amount of neurons. In this paper, we restricted the architecture to be as minimal as possible. As an illustration, if we happened to dropout at the latent layer, we may disconnect the network during the current gradient step ! Thus, dropout did not seem a good idea to us.
>
> 8/ size of 'd' is critical for autoencoders, only one example in appendix does not do justice, also it seems other color channels show up in the results (fig 10), wasn't it binary input?
>
> One of the main goals of this paper is to put ourselves in a situation where we know the optimal value of d (in this case d=1). The results referred to in the Appendix are from another work by Zhu et al. whose network attempts to learn the image manifold. The colour channels come from the fact that we used their code which is designed for colour images. We showed their results for d=1 and d=100 so that we could not be criticised for giving their algorithm too much or too little freedom in terms of dimensionality.

---

### Official Review · AnonReviewer1 · 2017-11-29
**a bit trivial and lacking in justification/insight as to the regularisation method**

**Rating:** 4
**Confidence:** 4

**Review:**

The paper considers a toy problem: the space of images of discs of variable radius - a one dimensional manifold.

An autoencoder based on convolutional layers with ReLU is experimented with, with a 1D embedding.

It is shown that
1) if the bias is not included, the resulting function is homogeneous (meaning f(ax)=af(x)), and so it fails because the 1D representation should be the radius, and the relationship from radius to image is more complex than a homogeneous function.
- if we include the bias and L2 regularise only the encoder weights, it works better in terms of interpolation for a limited data sample.

The thing is that 1) is trivial (the composition of homogeneous functions is homogeneous... so their proof is overly messy btw). Then, they continue by further analysing (see proposition 2) the solution for this case. Such analysis does not seem to shed much light on anything relevant, given that we know the autoencoder fails in this case due to the trivial proposition 1.

Another point: since the homogeneous function problem will not arise for other non-linearities (such as the sigmoid), the focus on the bias as the culprit seems arbitrary.

Then, the story about interpolation and regularisation is kind of orthogonal, and then is solved by an arbitrary regularisation scheme. The lesson learned from this case is basically the second last paragraph of section 3.2. In other words, it just works.

Since it's a toy problem anyway, the insights seem somewhat trivial.

On the plus side, such a toy problem seems like it might lead somewhere interesting. I'd like to see a similar setup but with a suite of toy problems. e.g. vary the aspect ratio of an oval (rather than a disc), vary the position, intensity, etc etc.

---

> ### Author Response · Authors · 2017-12-15
> **Reply to reviewer 1**
>
> 1/ It is shown that 1) if the bias is not included, the resulting function is homogeneous (meaning f(ax)=af(x)), and so it fails because the 1D representation should be the radius, and the relationship from radius to image is more complex than a homogeneous function.
> 2/ The thing is that 1) is trivial (the composition of homogeneous functions is homogeneous... so their proof is overly messy btw). Then, they continue by further analysing (see proposition 2) the solution for this case.
> 3/ Another point: since the homogeneous function problem will not arise for other non-linearities (such as the sigmoid), the focus on the bias as the culprit seems arbitrary.
>
> Yes, the fact that without biases the output of the AE is of the form alpha(r)F (where r summarises the input disk and F is a fixed image) is trivial, however it was necessary to state it. But we showed that in this case the AE actually finds the optimal possible F and function alpha(r), which is proportional to the dot product between the input disk and the function F, and not simply the area of the disk (see figures 7 and 8). The important fact here is that the ablated AE succeeds perfectly, in that it finds the best provable solution inside the range of its capacity. In deep learning it is rare to be able to show that the network is correctly approximating the best possible solution, therefore we found this point to be noteworthy. We address the question of using sigmoids in the reply to question 1/ of ``AnonReviewer2''.

---

### Official Review · AnonReviewer2 · 2017-12-03
**Interesting toy task but without real new insights or contribution to Autoencoders**

**Rating:** 4
**Confidence:** 3

**Review:**

This paper proposes a simple task (learning the manifold of all the images of disks) to study some properties of Autoencoders. They show that Autoencoders don't generalize to disks of radius not in the training set and propose several regularization to improve generalisation.

The task proposed in the paper is interesting but the study made is somewhat limited:

- They only studied one choice of Autoencoder architecture, and the results shown depends heavily on the choice of the activation, in particular sigmoid should not suffer from the same problem.

- It would be interesting to study the generalization in terms of the size of the gap.

- The regularization proposed is quite simple and already known, and other regularization have been proposed (e.g. dropout, ...). A more detailed comparison with all previous regularization scheme would be much needed.

- The choice of regularization at the end seems quite arbitrary, it works better on this example but it's not clear at all why, and if this choice would work for other tasks.

Also Denoising Autoencoders (Pascal et al.) should probably be mentioned in the previous work section, as they propose a solution to the regularization of Autoencoder.

Overall nothing really new was discovered or proposed, the lack of generalization of those kind of architecture is a well known problem and the regularization proposed was already known.

---

> ### Author Response · Authors · 2017-12-15
> **Reply to reviewer 2**
>
> 1/ They only studied one choice of Autoencoder architecture, and the results shown depends heavily on the choice of the activation, in particular sigmoid should not suffer from the same problem.
>
> The main idea of our paper is to study in detail the minimal generalisable architecture needed for encoding and decoding a disk, the rationale being that if the autoencoder does not work in that situation, then there is a serious problem. If it does work (which is indeed the case), then how does it work ? There is no reason that the problems discussed in our paper, such as generalisation, would be resolved using a sigmoid. We believe that this may be a misunderstanding due to the fact that we are learning with binary images. In such a case, it may indeed make sense to use a sigmoid, however this is not connected to the ablation study or the generalisation problem. We have carried out the same ablation study using sigmoids instead of leaky ReLUs and this did not resolve the observed behaviour.
>
> 2/ It would be interesting to study the generalization in terms of the size of the gap.
>
> This is indeed an interesting point, as clearly there should be some limit point at which the autonencoder cannot generalise. However, this would significantly increase the scope of the paper, making it too long for publication in the ICLR format.
>
> 3/ - The regularization proposed is quite simple and already known, and other regularization have been proposed (e.g. dropout, ...). A more detailed comparison with all previous regularization scheme would be much needed.
>
> We agree that regularising filter weights is widely known and used. However, we have not observed in the literature the asymmetric approach which consists in regularising only the encoder; if there are any such references, we would be happy if the reviewers could indicate them. In Section 3.2.4, we point out that regularising both the encoder and the decoder leads to a less stable autoencoder. We have carried out similar experiments on a 2D latent space (with ellipses), and found that this marked behaviour is observed again. We propose to add these experiments in our supplementary material, see
> https://www.dropbox.com/s/hn2akqgqh9m0qxg/autoencoders_sup_mat.pdf?dl=0
> Concerning comparison with other regularisation schemes, we cannot compare with sparse autoencoders, since these try to encourage sparsity in the latent space and in our case the latent space cannot be any more sparse.

---

### Author Response · Authors · 2017-12-15
**General reply to reviewers' comments**

We wish to thank the reviewers for their comments and criticisms, which we found to be useful and constructive. Before replying to the specific comments of the reviewers, we would first like to make some general points concerning the goal and scope of our work.

Firstly, we would like to clarify that in Section 3.2.2. we study a situation where we can describe the optimal solution of the training problem analytically. The ablation study was carried out and analysed to show that the autoencoder finds the optimal solution in this case. To the best of our knowledge, few such optimality results exist in the deep learning literature. Nevertheless, we do recognize that the case is simple.

Secondly, in Section 3.2.3. we investigate difficulties of the network to generalise. This is a very well-known problem concerning autoencoders (and GANs), which is often briefly discussed in the literature, but is rarely analysed in detail. In the case of complex images it is very difficult to decide whether a network is producing new examples or just copying examples from a database. We confirm the ubiquity of this problem by showing that it happens even in the case of the state-of-the-art work of Zhu et al. applied to disks.

Finally, in Section 3.2.4, we identify a solution to this problem in the form of an assymmetric weight regularisation, the regularisation of the encoder weights. This greatly improves the autoencoder's generalisation capacity. This asymmetric regularisation has not been proposed in the literature, to the best of our knowledge (please correct us if we are wrong in this respect). It is possible that in the submitted version of the paper we did not highlight this enough, but we believe it to be significant.

- New experiment : we have tested the asymmetric version of the regularization in a more complex case, with ellipses, and the improvement is even clearer than in the case of disks. We show these results in the following document
https://www.dropbox.com/s/hn2akqgqh9m0qxg/autoencoders_sup_mat.pdf?dl=0
Numerically, this leads to an order of magnitude improvement in the l2 loss of the network on unobserved examples.

Below are our replies to the specific comments of the reviewers.

---

### Decision · Program_Chairs · 2018-01-29
**ICLR 2018 Conference Acceptance Decision**

**Decision:**

Reject

**Comment:**

 + interesting approach for a detailed analysis of the limitations of autoencoders in solving a simple toy problem
 - resulting insights somewhat trivial, not really novel, nor practically useful => lacks demonstration of a gain on non-toy task
 - regularization study too limited in scope: lacking theoretical grounding, and more exhaustive comparison of regularization schemes.